# Challenges in Analyzing Functional Epigenetic Data in Perspective of Adolescent Psychiatric Health

**DOI:** 10.3390/ijms23105856

**Published:** 2022-05-23

**Authors:** Diana M. Manu, Jessica Mwinyi, Helgi B. Schiöth

**Affiliations:** Department of Surgical Sciences, Functional Pharmacology and Neuroscience, Uppsala University, 751 24 Uppsala, Sweden; jessica.mwinyi@neuro.uu.se (J.M.); helgi.schioth@neuro.uu.se (H.B.S.)

**Keywords:** epigenetic regulation, bioinformatics, statistical challenges, psychiatric conditions

## Abstract

The formative period of adolescence plays a crucial role in the development of skills and abilities for adulthood. Adolescents who are affected by mental health conditions are at risk of suicide and social and academic impairments. Gene–environment complementary contributions to the molecular mechanisms involved in psychiatric disorders have emphasized the need to analyze epigenetic marks such as DNA methylation (DNAm) and non-coding RNAs. However, the large and diverse bioinformatic and statistical methods, referring to the confounders of the statistical models, application of multiple-testing adjustment methods, questions regarding the correlation of DNAm across tissues, and sex-dependent differences in results, have raised challenges regarding the interpretation of the results. Based on the example of generalized anxiety disorder (GAD) and depressive disorder (MDD), we shed light on the current knowledge and usage of methodological tools in analyzing epigenetics. Statistical robustness is an essential prerequisite for a better understanding and interpretation of epigenetic modifications and helps to find novel targets for personalized therapeutics in psychiatric diseases.

## 1. Introduction

In the last decade, DNA methylation (DNAm) analyses have been extensively applied to uncover functional molecular mechanisms of diseases [1,2]. Changes in DNAm involve the addition of a methyl group to the C5 position of cytosine residues (5mC), primarily in CpG dinucleotides [3]. There are ~29 million CpGs in the human genome, and 60–80% of them are methylated [4]. High-throughput profiling methods (e.g., the Illumina EPIC microarray [5]) have made it easy to determine the DNAm in a large number of samples. However, robust statistical comparisons between differences in the DNAm and outcome or exposure should then be applied. Some critical issues that may affect the data analysis and outcome are the composition of the statistical models with confounders and/or covariates, choice of multiple-testing adjustment, how the sex distribution may drive the results, and how the tissue specificity of the DNAm can be translated into meaningful biological mechanisms for the target tissue. This review will scan through and address these challenges encountered when analyzing epigenetic modifications, with a focus on DNAm analyses in psychiatric disorders.

Worldwide, around 25% of the population is affected by psychiatric disorders. A broad definition of a good mental health is the one by the World Health Organization (WHO), defining it as “a state of well-being in which the individual realizes his or her own abilities, can cope with the normal stresses of life, can work productively and fruitfully, and is able to make a contribution to his or her community” [6]. Generalized anxiety disorder (GAD) and major depression disorder (MDD) represent the most prevalent mental disorders that may lead to negative consequences, such as alterations in intellectual ability, behavior, affectivity, and social relations. Moreover, they represent a high risk for a number of other physical conditions, including diabetes, hypertension, obesity, and suicide, especially at a young age (Figure 1). These observations highlight the need for the early recognition and treatment of mental disorders.

The recognition that DNAm plays a crucial role in psychiatric disorders has led to a plenitude of statistical models and approaches in the field to be able to evaluate and interpret the immense amount of data produced in genome-wide experiments. Age, sex, genetics, medication, environmental exposures (notably smoking status [7]), and cellular composition of the investigated tissue were demonstrated to have an important effect on the DNAm and should be included in the statistical model. However, the inclusion of too many confounders/covariates may result in an overfit of the statistical model. A rigorous method for multiple-testing comparisons should be further applied to the statistical results that is suitable for the analyzed sample size. Moreover, most of the methylation studies have been using accessible tissues such as whole blood and saliva to infer the mechanisms related to psychiatric outcomes [8,9,10,11,12,13,14,15,16].

The results may be mechanistically informative by making use of the available bioinformatic tools and publicly available data from the consortia and other research groups. This review offers an updated synthesis and knowledge on the main bioinformatics and biostatistics challenges a researcher may be confronted with in order to make sense of the epigenetic data in the context of psychiatric disorders.

### 1.1. Prevalence and Risk Conditions of MDD and GAD

Depression and GAD cooccur in three out of five cases, being highly reported in primary care and emergency settings [20]. Depression is the most prevalent psychiatric disorder, affecting 6% of the adult population each year [21], and almost 40% experience the first episode of depression before the age of 20 [22]. GAD has a lifetime prevalence of 2.8–4.1% in Europe and the US [17,18]. Importantly, depression is responsible for more "years lost” to a disability than any other condition [23]. The substantial personal cost of depression is estimated to be up to 50% of the 800,000 suicides that occur per year [24], and individuals with MDD are almost 20-fold more likely to commit suicide than the general population [25]. Both depression and GAD are associated with an increased risk for chronic physical conditions, such as asthma, arthritis, back/neck problems, chronic headache, diabetes, heart disease, hypertension, multiple pains, obesity, and ulcers, and the risk significantly increases when MDD and GAD co-occur (Figure 2) [26]. Prompt recognition and effective treatment against GAD and depression are, thus, needed.

### 1.2. DNAm as Gene–Environment Interplay in Depression and GAD

DNAm is a high-fidelity manifestation that reflects the interdependence of genes and the environment. The DNAm levels either change rapidly, with shifts that occur within 1 h and reverse by 24 h [27], or they are stable, containing events from prior decades [28]. DNAm modification is induced by developmental and exogenous factors, such as diet [28], cigarette smoking [29], age [30], medication [31], and physical activity [32]. Smoking during prenatal pregnancy has consequences of widespread and highly reproducible differences in the DNAm at birth [33]. Medication administrated against depression may interact with epigenetic mechanisms to create a “window of synaptic plasticity” [34]. This may further lead to interindividual differences in response to antidepressants through differential DNAm and enzyme mRNA expression [35]. DNAm is also an attractive candidate to explain sex-dependent differences in brain functions and vulnerability to diseases [36,37,38]. Studies on fetal brains [39,40] have inferred that sex differences in the brain methylome occur mostly early on in fetal development and are stable throughout life. Similarly, increased *NR3C1* DNAm levels were observed in girls when their mothers reported lower maternal depression scores during pregnancy, while no evidence of such a relation was observed in boys [41]. Lastly, besides the beneficial effects of breastfeeding on newborn infants, studies have reported that breastfeeding mothers report reductions in anxiety, negative mood, and stress when compared to formula-feeding mothers [42].

Twin studies confirmed interindividual variations across tissue types, suggesting that the DNA sequence has an effect on the DNAm levels [43]. Genetic backgrounds may account for nearly 20–80% of the DNAm variance at certain sites known as methylation quantitative trait loci (meQTLs) [44,45]. In whole blood, the heritability of DNAm is around 19% across the epigenome [45], of which ~7% is captured by common genetic variants [46]. To this end, single-nucleotide polymorphisms (SNPs) with a cis-meQTL effect (defined as variants <1 Mb from the DNAm site) lead to variations in the DNAm status and often cooccur with the expression of QTL (eQTL) or other regulatory QTLs [47,48,49]. Importantly, the genetic effects on DNAm are stable over the life course [50] and across tissues, suggesting that some meQTLs could exert ubiquitous effects on DNAm [51]. Based on the genetic and epigenetic data of >30,000 participants, an online resource is available to examine cis- and trans-meQTLs at http://mqtldb.godmc.org.uk/ (accessed on 11 May 2022) [52].

Multiple studies have investigated DNAm in depression and GAD, but the findings are not consistent across the investigations that detected different susceptibility targets. Looking at brain-derived neurotrophic factor (BDNF), a critical mediator of neuronal activity [53], which has been linked to depression [54], researchers found increased DNAm mainly in the promotor of exons I and IV in depressed patients [55,56,57,58]. Other studies focused on SLC6A4 [59,60,61] and glucocorticoid receptor NR3C1 [62,63,64], but the results were not in the same direction. A recent study on DNAm at KLK8 found an association with the severity of depression symptomology, as investigated in 80 depression cases and 80 gender-matched controls [65]. A meta-analysis, including 11,256 middle-aged and elderly multiethnic people, found associations of the DNAm levels and depressive symptoms at three CpG sites, implicating the genes involved in axon guidance [66]. In addition, using 150 monozygotic twin pairs discordant for early onset MDD, the study identified 760 genomic sites that were differentially methylated, enriching for neuronal circuitry and plasticity genes [67]. Lastly, the meQTL effect was demonstrated by Xueyi Shen et al. in peripheral blood and, further, the mutual causal effect of DNAm on the liability of MDD at the associated CpG probes [68]. To our knowledge, two studies investigated the relationship between DNAm and GAD. A targeted analysis of DNAm at NR3C1 revealed higher blood DNAm levels in patients with GAD [69]. Ciuculete et al. validated the DNAm shifts at the STK32B gene in relationship with the GAD risk in adolescence [13]. Other studies focused on a whole group with anxiety and identified higher global DNAm and at the ASB1 gene promoter in anxious individuals compared to non-anxious controls [70,71]. Importantly, DNAm may also affect the transcription of miRNA genes, which are part of the epigenetic regulatory network. miRNAs are short DNA-transcribed molecules highly expressed in the brain that modulate gene expression at the post-transcriptional level [72]. In a recent review on postmortem brain samples of depressed subjects, approximately 50 miRNAs were identified to be differentially expressed compared to the healthy counterparts; however, miRNA-124 was consistently reported (see Review [73]). In blood, miRNA-134 was significantly downregulated in depressed patients and in rats affected by mild stress [74]. A study investigating the miRNA levels in patients with GAD identified that the circulating miRNA-4505 and miRNA-663 levels correlated with the symptoms of anxiety [75]. Lastly, miRNA-144 was found to have a lower expression in patients with MDD and anxiety [76], and differential expression at cluster miRNA-17-92 was observed in mice with elevated anxiety- and depression-like behaviors [77].

### 1.3. Methodological Challenges in DNAm Analysis in Psychiatric Disorders

#### 1.3.1. The Choice of the Investigated Tissue

The choice of the analyzed tissue as a substitute for the brain is important, as patterns of epigenetic covariation are tissue-specific. Several studies investigated epigenetics using postmortem brain samples, resulting in inconsistent findings [78,79,80]. Potential sources of confounding when analyzing postmortem cortical tissue include a terminal condition or the agonal state (that is, the condition of the samples at death), cause of death, postmortem interval (PMI), and tissue dissection and processing [81,82]. The PMI influence was not only shown to predict the amount of DNA degradation, but it seems to also play a role in the fidelity of the measured DNAm data [83]. A lower pH of the brain samples may affect the RNA and protein integrity; however, it does not seem to have an effect on the methylation marks [81]. Last, but not at least, distinct epigenetic landscapes were detected among diverse brain regions [84], possibly driven by the neuronal cell population [85]. A common and easily accessible proxy for the brain used in the majority of studies investigating psychiatric disorders is whole blood [8,10,11,86]. Additionally, another advantage of using a proxy tissue is that the sample size is increased and the penalty of multiple-testing comparisons of DNAm at around 850,000 sites is overcome [87]. Moreover, psychiatric disorders are characterized by chronic inflammation, exhibiting an increase in immune molecules, including inflammatory cytokines and acute phase proteins [88]. Studying DNAm signatures in white blood cells may be used as clinically useful biomarkers. Although the central nervous system may have a different inflammatory response than whole blood as a consequence of the environmental upset [89], this may particularly change in the case of a dysfunction of the blood–brain barrier [90]. Three hypotheses were stated by M. Szyf regarding the association between DNAm in the blood and behaviors or brain-related phenotypes [91]. The first one describes how DNAm changes observed in the blood do not reflect what is happening in the brain, as they occur as a consequence of environmental upset. Second, if an environmental upset happens early in life, it may impact common precursor cells to the blood and brain, causing similar DNAm changes. Lastly, circulating molecules such as hormones, insulin, and miRNAs, which are released in response to an external factor, will cause the same DNAm changes in the blood and brain. It is likely that DNAm changes observed in the blood represent a combination of the postulated hypotheses, and thus, investigating the blood–brain relationship is crucial in providing biological relevance and mechanisms.

Several studies have investigated how DNAm shifts in whole blood may be biologically informative for shifts that may occur in the brain and, thus, brain function. Walton et al. inferred that 7.9% of the total CpG sites had a strong correlation regarding the DNAm levels, using blood and temporal lobe biopsy samples from 12 epilepsy patients [92]. Looking at the most variable CpG probes in the blood (*n* = 185,060 CpGs), Hannon E. et al. found an overrepresentation of correlated DNAm patterns between whole blood and the brain at loci in the vicinity of the transcription start site, first exon, and 5′ untranslated region using a large sample of 122 individuals [93]. A searchable online database was developed by Hannon E. et al. http://epigenetics.essex.ac.uk/bloodbrain/ (accessed on 11 May 2022). In a more recent study, Edgar et al. investigated the concordance in DNAm between the blood and three brain regions, i.e., Brodmann Area (BA) 7, BA10, and BA20, of 16 individuals. The research group detected that 9.7% of CpGs correlated in DNAm between the blood and any of the three brain regions. A web application named BECon (https://redgar598.shinyapps.io/BECon/, accessed on 11 May 2022) allows the analysis on the variability of cytosine-phosphate-guanine dinucleotides (CpGs) in blood and brain samples in concordance with the determination of CpGs between the blood and brain and estimations of how strongly a CpG is affected by cell composition in both the blood and brain. Comparing the study findings across the last two mentioned studies, they found that the overlap between the lists of informative CpGs regarding DNAm was greater than expected by chance. Apart from whole blood, saliva and buccal samples also represent attractive substitutes for brain tissue. A total of 21 samples of live brain, saliva, blood, and buccal were collected and analyzed by the Illumina EPIC array. When the DNAm levels were averaged across all individuals, a higher correlation was observed between the brain and saliva tissues, similar to previous studies [94,95]. However, the cross-tissue correlations were strongest between the blood and brain when looking at individual CpGs within a subject [95]. A possible rationale behind these observations is that the DNAm correlations may be specific to an individual tissue cell/region or, at the very least, have different magnitudes of effects across the cell types and tissues as a consequence of a given factor (e.g., environment, disease, or genotype).

#### 1.3.2. Available Tools for the Biological Interpretation of DNAm Findings

Various online tools and databases can help to infer biological and mechanistic interpretations of DNAm changes. The Encyclopedia of DNA Elements (ENCODE) project has developed a well-applicable functional annotation on chromatin organization of the human genome [96]. Applying multivariate hidden Markov models on a variety of cell lines, the genome was divided into 15 chromatin states [97,98]. This method is advantageous for capturing regulatory elements with great reliability, robustness, and precision [97]. The available chromatin states collected in ENCODE correspond to repressed, poised, and active promoters; strong and weak enhancers; putative insulators; transcribed regions; and large-scale repressed and inactive domains. In the studies investigating psychiatric disorders in proxy tissues, the chromatin states increase the knowledge on the regulatory role of the differentially methylated genomic sites in peripheral blood mononuclear primary cells and various regions of the brain, e.g., hippocampus, cerebellum, substantia nigra, and angular gyrus. The chromatin modification through enzymatic activity and DNAm may modulate the chromatin structure of histones in adjacent nucleosomes or the interaction of histones with DNA [99]. While some studies claim that DNAm makes the DNA less flexible and less likely to form nucleosomes [100,101], others have reported that DNAm increases the affinity of histones for DNA [102] and DNAm facilitates compaction on preassembled nucleosomes [103]. The most known chromatin modification is acetylation, which facilitates the unfolding of chromatin by neutralizing the basic charge of the lysine. The status of the net histone acetylation status determined by the rate between histone acetyltransferase enzymes (HATs) and histone deacetylases (HDACs) plays an important role in chronic stress and in the responses to antidepressant therapy [104]. Several studies on animal models have reported a low activity of histone acetylation in the nucleus accumbens of mice [105], while the reduction of HDAC has had an antidepressant effect in animals with stress-induced depressive-like behaviors [106,107]. In postmortem brains of the depressed subjects, H3K14ac was elevated, and HDAC2 was decreased [105], and the expression levels of HDAC2 and HDAC5 mRNAs in the peripheral white blood cells were elevated in depressed patients compared to healthy controls [108]. However, while the methodological evaluation of this review focused on DNAm, similar evaluations of relevant aspects of other epigenetic regulatory mechanisms should be covered in a separate review.

In addition, ENCODE provides information on long-range interactions in four cell types (GM12878, K562, HeLa-S3, and H1 hESC), uncovering distal functional regulatory mechanisms in a disease-relevant cell context. For instance, an enhancer and its distal gene target are located close to each other in the three-dimensional space through long-range chromatin interactions, i.e., enhancer–promoter interactions [109]. Other consortia have contributed with relevant data to the ENCODE project. For example, the NIH Roadmap Epigenomics Program, which provided information about histone marks, DNAm, DNA accessibility, and RNA expression in order to infer high-resolution maps of regulatory elements annotated jointly across a total of 127 reference epigenomes spanning diverse cell and tissue types [110]. The WashU Epigenome Browser http://epigenomegateway.wustl.edu/browser/ (accessed on 11 May 2022) is an interactive tool for genomic data exploration, including chromatin analysis, long-range interactions, and epigenetic marks [111]. The FANTOM consortium has applied a capped analysis of the gene expression (CAGE) methodology to examine the transcription initiation and promoter usage for a variety of cell and tissue types [112]. The GTEx project provides transcriptome data for most tissue types. The cumulative work of the consortia and laboratories has helped to produce a greatly enhanced view of the human genome and the regulatory roles at many genomic loci and genes [98]. Cross-checking data generated by the ENCODE project, GTEx, FANTOM5, and context- and tissue-dependent effects may be observed.

#### 1.3.3. Validation of the DNAm Findings by a Different Technique or in an Independent Cohort

The advancement made in experimental profiling techniques and computational analysis approaches have led to a huge amount of DNAm data that is accumulated and needs to be analyzed. Three approaches are well-studied for DNAm interrogation, i.e., the bisulfite conversion-based, methylation sensitive enzyme restriction-based (MSRE), and affinity enrichment-based methods [113]. During the MSRE and affinity-based approaches, the DNA is not damaged by the bisulfite treatment; however, they require more labor in retrieving the DNA fragments bound by the respective proteins. Furthermore, the interpretation of the DNAm data is somewhat challenging for this type of data. Following bisulfite conversion, the degree of DNAm is quantified by microarray or next-generation sequencing (NGS). By far, the most popular approach is the analysis of a preselected set of CpGs via DNA hybridization microarrays. The array-based technique is mostly used for the screening of DNAm profiles across more than 850,000 CpG sites. In addition, reproducibility has been shown for different arrays, which makes it possible to compare previous results with new ones [114]. Although it has the disadvantage that the probes are preselected, leaving a large proportion of CpG sites unmeasured, microarray techniques have been shown to measure small methylation differences (0.5–10%), which are common in other conditions than cancer [115,116]. Differences of, e.g., 0.5–1.1% were identified between type 2 diabetes and controls measured in whole blood by the Illumina 450 K BeadChip [116]. Pyrosequencing is advantageous in terms of the DNAm assessment of a specific locus and an accurate base resolution [117]. The disadvantages are, however, a relatively high experimental effort, a relatively high price, and missing possibilities to correct for cellular heterogeneity, as needed for determinations performed in blood cells [118]. Pyrosequencing has been efficiently used as a validation method of the DNAm changes measured by high-throughput methods at specific sites [9,119,120]. The sensitivity limit of the pyrosequencing methods lies around 5% [121]. The approaches based on bisulfite conversion do not distinguish between 5mC and 5-hydroxymethl cytosine modifications. This drawback is especially important in studies of DNAm in psychiatric disorders, as there is evidence of abundant tissue-specific stable hyroxymethylation in neurons [122]. In comparison to microarrays, which do not require any read alignments, sequence-based processing includes the trimming of unwanted bases from the reads, such as sequencing adapters, or unwanted bases resulting from enzymatic end repair. Once those trimmed sequencing reads are aligned to the reference genome, methylation is started. Importantly, the sequencing methods provide a good resolution of genome-wide methylation, allowing to explore methylation patterns far beyond the single-site methylations shown by arrays [123].

A straightforward strategy to draw biological and functional consequences is to correlate DNAm with the mRNA gene expression. In addition, plenty of computational tools are available, such as the R package BioMethyl, which allows to integrate several algorithms for the interpretation of DNAm data [124]. Another software, named EpiExplorer, provides the interactive and live exploration of differentially methylated sites in the context of public reference epigenome datasets [125]. The Galaxy software facilitates the comparison of data at the genomic regions with other genomic datasets that are available online [126]. A gene set enrichment analysis can be carried out using the GREAT Web server [127] which maps genomic regions to genes and controls the statistical analysis for the fact that genes differ in size and in their relative distance to each other. The interpretation of the DNAm shifts integrated with other omics data, such as ChIP-seq, remains difficult, both technically and biologically.

Although less sensitive, an additional way to confirm DNAm findings is the usage of multiple independent cohorts [65]. A large depository of open-access data of DNAm is available through the database ArrayExpress (https://www.ebi.ac.uk/arrayexpress/, accessed on 11 May 2022). When using an independent cohort for DNAm confirmation, it should be noted that the validation cohort may be characterized by different ages, sex, environmental exposure, and medication usage compared to the discovery cohort. These aspects may have an effect on the DNAm levels. Moreover, some variables, such as, e.g., cell estimations, may not be available for the deposited dataset. A strength of the open-access depositories is the diversity of the DNAm levels in various tissues, making it possible to confirm the DNAm findings in brain. The validation of the findings in another group of people should, thus, increase the applicability and biological relevance of the DNAm differences.

#### 1.3.4. Composition of the Statistical Models when Analyzing DNAm and Statistical Significance

The statistical model’s composition is different across the studies looking at DNAm, potentially leading to inconsistent results. Age and sex are two essential confounders, together with the batch and the tissue cell-type proportions, which are believed to give the highest variability [128,129]. In the case of psychiatric disorders, DNAm analyses should be performed separately in men and women or, if the sample size allows, an interaction term of sex–disease considered as a covariate. In order to avoid an overfit of the model with too many covariates, principal components (PCs) are good candidates for covariates in the statistical model in the case of moderate sample sizes. PCs take the variability from various sources into account, including the genetic source. Due to the known interactions between genetic and epigenetic factors, the inclusion of genetic risk models should always be considered when DNAm data are statically analyzed. McCartney et al. showed that the DNAm scores explained 61% of the variance in the smoking status and 12.5% of the variance in the BMI and alcohol consumption and contributed additively when the polygenic risk score was included in the analysis (Figure 3) [130].

Another aspect that may sometimes confuse a researcher when reading different articles is the use of different thresholds to report a significance. A nominal *p*-value of 0.05 or a Bonferroni, FDR, or BH adjustment for multiple testing may be used for statistical comparisons. If the number of hypotheses cannot be reduced, then the likelihood for both false-positive and false-negative conclusions should be kept at a low rate. The most stringent adjustment is the Bonferroni method, which controls the proportion of false positives, but a true relation may go unnoticed (type 2 error). In general, it is recommended to perform sample size calculations, including the Bonferroni correction, before the analysis is started [131]. A web app that has been developed to perform power calculations for DNAm is found at https://epigenetics.essex.ac.uk/shiny/EPICDNAmPowerCalcs/ (accessed on 11 May 2022). For DNAm data, where the sample size and the expected effect size are usually moderate, the false discovery rate (FDR) proposed by Benjamin and Hochberg [132] represents an appealing solution. The two popular FDR-based procedures for genome-wide multiple testing are the original BH procedure [132] and Storey’s q value procedure [133]. Compared to the BH procedure, the latter approach considers the proportion of null hypotheses and is more powerful when the signal is dense [134]. Independent of the number of comparisons included in the epigenome-wide analysis, Mansell et al. proposed a significance threshold of 9 × 10^−8^ for the EPIC array [87], and Saffari and colleagues estimated a *p*-value of 2.4 × 10^−7^ for the 450 K array [135]. However, these significance thresholds can be considered as still too conservative, and researchers tend to report DNAm results at a more lenient threshold, using the BH and FDR procedures [136,137,138,139,140,141].

#### 1.3.5. Adjusting for Cell-Type Proportions in Whole Blood

Whole blood represents an aggregate of different cell types. The proportions of whole blood cell types may change due to external factors, such as, e.g., smoking [142], age [143], and food [144]. Unlike germline genetic variations, DNAm signatures are cell type-specific [119]. The DNAm levels measured in the whole blood should be seen as the sum of DNAm at the constituent cell types weighted by their abundance. A number of reference-based and reference-free cell-type deconvolution algorithms have been proposed to adjust for cell type-specific epigenetic alterations. The calculated deconvolution estimates can be further used in statistical models as covariates [145] or investigated separately to determine their association with risk or exposure [146]. The most used method of the calculation of blood cell proportions is the reference-based Houseman algorithm, implemented in the *minfi* R package [15,137,147,148]. The reference data were from six adult White European males. Given that DNAm patterns vary by sex, age, and ancestry, the algorithm should be used with caution in non-European populations of mixed ages and sexes. A PC analysis on a subset of informative and highly variable CpG sites may also be a method to account for the cell proportions of individual samples [149]. This method should, however, be applied with caution when a dense signal is expected, as it has the potential problem of overcorrection and, hence, loss of power [150].

#### 1.3.6. Interpretation of DNAm Variation, SNPs, and miRNA as a Complex Network

The limited direct contribution of SNPs to the susceptibility of complex traits has led to the need to analyze the genetic variants in the context of their effects on regulatory entities and mechanisms, such as DNAm, TFs, and miRNAs (Figure 4). The meQTL effect may take place both locally [44] and globally through the three-dimensional reorganization of chromatin. The three-dimensional structure of DNA can implicate physical contact between distal genomic locations or even discrete chromosomes by protein-mediated looping [151]. A long-range interaction analysis allows collecting knowledge on the contact between distal and proximal regulatory elements and gene promoter sites, which can together alter the mRNA expression [152]. Earlier studies estimated that 15% of CpGs were under genetic control [153], and recently, the estimation increased to 55% [129]. Moreover, meQTLs are, in general, enriched for the active chromatin states [153] and may impact alternative splicing [154]. The interplay of SNPs with epigenetic factors such as miRNAs and DNAm results in a preferential mRNA expression of one gene allele, thereby leading to an allele-specific gene expression (Figure 4) and variations in mRNA expression output [155]. Importantly, the SNP–epigenetic interaction is implicated in interindividual differences regarding the susceptibility for human diseases and drug response [156]. This interaction results in differential effects, with some individuals more vulnerable to environmental adversity. Studies suggest referring to genes as “plasticity genes”, as, depending on their individual genetic backgrounds, the gene response may be equally positive or negative to environmental influences [157]. Last, but not least, more attention should be given to the fact that epigenetic mechanisms are not independent but work together and cross-regulate each other in a common regulatory network [152,158].

## 2. Concluding Remarks

The crucial role of DNAm in the pathophysiology of various diseases has led to a revolution in the analytic approaches. The challenges in data analysis, which are addressed differently by the researchers, have partly resulted in inconsistent and even contradictory findings. Besides the complexity and timing/dynamicity of the interdependence of the environmental factors with the genetic background, an optimal choice of the statistic approach may help with a more robust and convergent interpretation of the epigenetic results. Although the knowledge regarding the important confounders of DNAm has grown over the time, such as the huge contribution of sex, age, batch, cell-type proportions, medication, and genetics, the difficulty in incorporating these factors into the statistical models to evaluate the DNAm shifts and differences remains. While some covariates, such as age and sex, should be self-included in the statistical model, PCs represent a convenient way to adjust for variances induced by multiple sources without an overfit of the model. Studies whose aim was to identify DNAm shifts as biomarkers, then easily accessible tissues, which show good sensitivity and specificity for patients at risk, are the main requirement. However, if the aim of the study is to increase the knowledge in mechanisms underlying psychiatric phenotypic susceptibility, the DNAm results should be further investigated with regards to their possible implications in the pathophysiology of the disease. Conclusions regarding cross-tissue mechanisms can be inferred only after a comprehensive analysis of the DNAm results in various tissue and cell-type contexts, suggesting that interindividual DNAm variations due to the investigated traits are higher than interindividual DNAm differences from other sources. At present, there are many online tools publicly available that provide DNAm data from various tissues that can be used to assess the tissue specificity of the DNAm findings (Table 1). Lastly, DNAm results can be integrated into the complex regulatory network by downstream investigation of the effect of DNAm on the mRNA gene and/or non-coding RNA expression levels. In light of the major statistical challenges, it would be important that the applied statistical methods and functional characterizations will be addressed in a consensus by researchers. The future of epigenetic knowledge implicates exciting and impactful discoveries and customized treatments ahead.

## Figures and Tables

**Figure 1 ijms-23-05856-f001:**
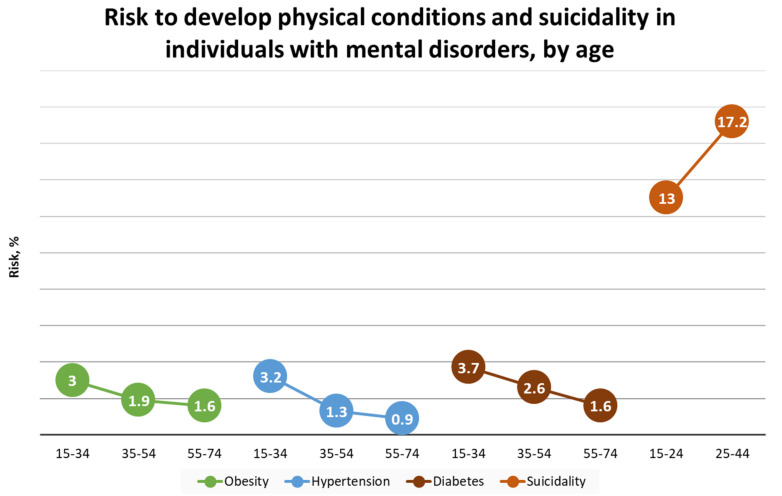
Risk of developing poor physical conditions and death from unnatural causes in individuals with mental disorders by age. Data on obesity, hypertension, and diabetes were extracted from The Health Improvement Network in 2018 in England, comprising 1,051,127 patients. Out of them, 9357 (0.9%) had a diagnosis of severe mental illness (SMI). The rate ratios of the prevalence were calculated between the SMI and all patients in three age groups, i.e., 15–24, 25–54, and 55–74 [17]. The standardized mortality ratio from suicide was calculated using 1,003,906 patients with hospital discharge with depression from a dataset of English national hospital episode statistics, linked with data from death records. The ratios were calculated comparing mortality in people with depression with mortality in the general population of England in two age groups: 15–24 and 25–44 years [18]. Reprinted/adapted with permission from ref. [19]. 2022, Diana M. Manu.

**Figure 2 ijms-23-05856-f002:**
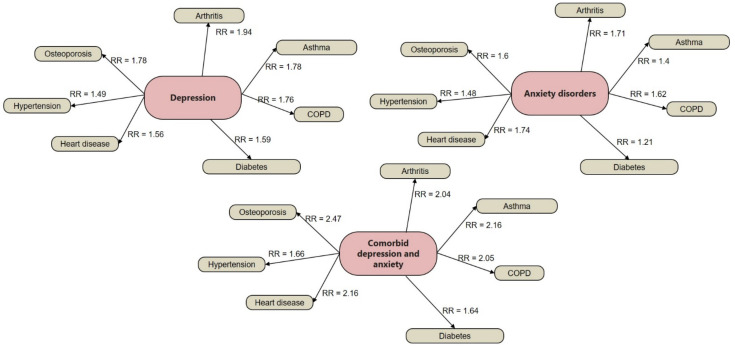
Associations between depression, anxiety, and comorbid depression–anxiety and chronic physical condition risk ratios (RR). Analytic sample comprised of 33,242 adults (51% women) with ages between 22 and 64 years, with no self-reported diagnosis of schizophrenia, psychoses, attention deficit hyper activity disorders, adjustment disorders, substance abuse disorders, and personality disorders. COPD = Chronic Obstructive Pulmonary Disease. Individuals were part of the Medical Expenditure Panel Survey annual releases of 2007 and 2009 in the United States. Data were published in [26]. Reprinted/adapted with permission from ref. [19]. 2022, Diana M. Manu.

**Figure 3 ijms-23-05856-f003:**
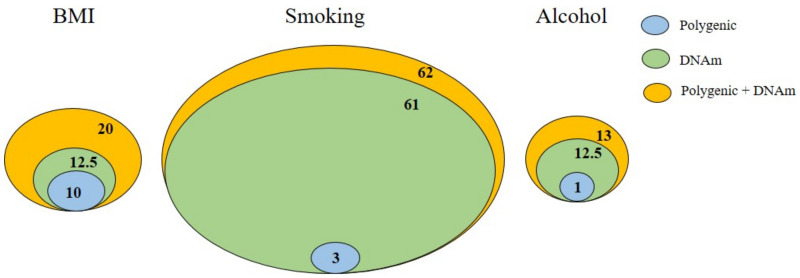
DNAm and polygenic prediction of the health and lifestyle factors. The proportion of phenotypic variance explained (R2) is plotted for three traits: the body mass index (BMI), smoking, and alcohol consumption (alcohol), based on each trait’s polygenic score (blue), DNA methylation-based score (green), and additive genetic + epigenetic score (yellow). Data were taken from [130]. Reprinted/adapted with permission from ref. [19]. 2022, Diana M. Manu.

**Figure 4 ijms-23-05856-f004:**
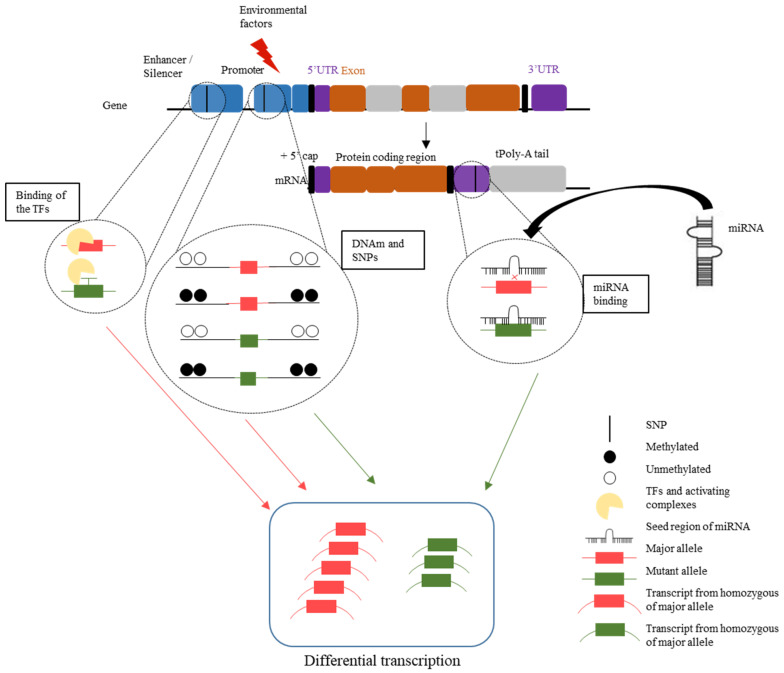
The interplay of SNPs, TFs, DNAm, and miRNA binding, leading to differential mRNA gene expression. SNPs, single-nucleotide polymorphisms; TFs, transcription factors; DNAm, DNA methylation; miRNA, microRNA. Reprinted/adapted with permission from ref. [19]. 2022, Diana M. Manu.

**Table 1 ijms-23-05856-t001:** Web-based tools for an epigenetic analysis.

Website (accessed on 11 May 2022)	Developer	Query	Reference
http://mqtldb.godmc.org.uk/	Josine L Min et al.	DNAm-meQTL	[52]
http://epigenetics.essex.ac.uk/bloodbrain/	Hannon E. et al.	Blood–brain DNAm correlation	[93]
https://redgar598.shinyapps.io/BECon/	Edgar et al.	Blood–brain DNAm correlation	[159]
http://epigenomegateway.wustl.edu/browser/	Xin Zhou et al.	Exploration of genomic data	[111]
https://www.ebi.ac.uk/arrayexpress/	Alvis Brazma et al.	Open-access data	[160]
https://epigenetics.essex.ac.uk/shiny/EPICDNAmPowerCalcs/	Mansell et al.	Power calculation	[87]

## Data Availability

Not applicable.

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
