# Peer review of "Challenges in Analyzing Functional Epigenetic Data in Perspective of Adolescent Psychiatric Health"

_ijms, 2022, doi:10.3390/ijms23105856_

Round 1

Reviewer 1 Report

In this review, Manu and colleagues have attempted to describe the potential challenges associated with the analysis of DNA methylation studies. The review is well written, and the authors' major points clearly conveyed. I have the following comments:

  1. In the section describing the choice of tissue for the DNAm studies, it will be useful for the general reader if the authors have provided more information about the advantages and disadvantages of using different tissues to assess DNAm concerning GAD MDD.  
  2. The section describing the tools for explaining DNAm can be expanded further by actually providing a discussion on how other available datasets explain DNAm. It will be helpful to have a discussion on how for example, chromatin changes (that itself is subjected to differential methylation) impact DNA methylation and disease initiation/progression. 
  3. In addition to the Illumina array-based DNAm assays, there are also NGS based approaches for assessing DNAm on a genome-wide level, such as MBD capture, followed by sequencing. Discussion on the advantages and disadvantages between the array and NGS platforms for assessing DNAm would benefit the readers.  

Author Response

In the section describing the choice of tissue for the DNAm studies, it will be useful for the general reader if the authors have provided more information about the advantages and disadvantages of using different tissues to assess DNAm concerning GAD MDD.” 

Thank you for the comment. We have now added further discussion in the paragraph “1.3.1. The choice of the investigated tissue”. “[…] Several studies investigated epigenetics using postmortem brain samples, resulting in inconsistent findings (1-3). Potential sources of confounding when analyzing post-mortem cortical tissue include terminal condition or the agonal state (that is, the condition of the samples at death), cause of death, postmortem interval (PMI) and tissue dissection and processing (4, 5). PMI influence was not only shown to predict the amount of DNA degradation, but it seems to also play a role in the fidelity of the measured DNAm data (6). A lower pH of the brain samples may affect RNA and protein integrity, however, it does not seem to have an effect on methylation marks (4). Last, but not at least, distinct epigenetic landscapes were detected among diverse brain regions (7), possibly driven by the neuronal cell population (8). […] Moreover, psychiatric disorders are characterized by chronic inflammation, exhibiting increase in immune molecules, including inflammatory cytokines and acute phase proteins (9). Studying DNAm signatures at white blood cells may be used as clinically useful biomarkers. Although the central nervous system may have a different inflammatory response than whole blood as a consequence of the environmental insult (10), this may particularly change in case of a dysfunction of blood-brain barrier (11). Three hypotheses were stated by M. Szyf regarding the association between DNAm in blood and behavior or brain-related phenotypes (12). The first one describes how DNAm changes observed in blood do not reflect what is happening in brain as they occur as a consequence to environmental insult. Second, if an environmental insult happens early in life, it may impact common precursor cells to blood and brain, causing similar DNAm changes in blood and brain. Lastly, circulating molecules such as hormones, insulin and miRNAs which are released in response to an external factor will cause same DNAm changes in blood and brain. It is likely that DNAm changes observed in blood represent a combination of the postulated hypotheses and thus, investigating the relationship blood-brain is crucial in providing biological relevance and mechanisms.

“The section describing the tools for explaining DNAm can be expanded further by actually providing a discussion on how other available datasets explain DNAm. It will be helpful to have a discussion on how for example, chromatin changes (that itself is subjected to differential methylation) impact DNA methylation and disease initiation/progression.”

We thank the reviewer for this comment. We have now added a summarized discussion in the paragraph “1.3.2. Available tools for biological interpretation of DNAm findings” on how chromatin change may be modulated by depression and DNAm. “The chromatin modification through enzymatic activity and DNAm may modulate the chromatin structure of histones in adjacent nucleosomes or the interaction of histone with DNA (13). While some studies claim that DNAm makes the DNA less flexible and less likely to form nucleosomes (14, 15), others have reported that DNAm increases the affinity of histones for DNA (16) and DNAm facilitates compaction on pre-assembled nucleosomes (17). The most known chromatin modification is acetylation which facilitates the unfolding of chromatin by neutralizing the basic charge of the lysine. The status of the net histone acetylation status determined by the rate between histone acetyltransferase enzymes (HATs) and histone deacetylases (HDACs) play an important role in chronic stress and responses to antidepressant therapy (18). Several studies on animal models have reported low activity of histone acetylation in the nucleus accumbens of mice (19), while reduction of HDAC had an anti-depressant effect in animals with stress-induced depressive-like behaviors (20, 21). In postmortem brain of depressed subjects, H3K14ac was elevated, and HDAC2 was decreased (19), and the expression levels of HDAC2 and HDAC5 mRNAs in peripheral white blood cells were elevated in depressed patients compared to healthy controls (22). However, while the methodological evaluation made in this review focuses on DNAm, similar evaluation of relevant aspects of other epigenetic regulatory mechanism should be covered in a separate review.

In addition to the Illumina array-based DNAm assays, there are also NGS based approaches for assessing DNAm on a genome-wide level, such as MBD capture, followed by sequencing. Discussion on the advantages and disadvantages between the array and NGS platforms for assessing DNAm would benefit the readers.

Thank you for this comment. We have now added this in the paragraph “1.3.3. Validation of the DNAm findings by a different technique or in an independent cohort”. Three approaches are well studied for DNAm interrogation, i.e. bisulfite conversion-based, methylation-sensitive-enzyme-restriction-based (MSRE) or affinity enrichment-based methods (23). During the MSRE and affinity-based approaches, DNA is not damaged by bisulfite treatment, however, they require more labor in retrieving the DNA fragments bound by the respective proteins. Furthermore, the interpretation of the DNAm data is somewhat challenging for this type of data. Following bisulfite conversion, the degree of DNAm is quantified by microarray or Next Generation Sequencing (NGS). By far, the most popular approach is the analysis of a preselected set of CpGs via DNA hybridization microarrays. […] In addition, reproducibility has been shown for different arrays, which makes it possible to compare previous results with new ones (24). […] The approaches based on bisulfite conversion do not distinguish between 5mC and 5-hydroxymethl cytosine modifications. This drawback is especially important in studies of DNAm in psychiatric disorders, as there is evidence for abundant tissue specific stable hyroxymethylation in neurons (25). In comparison to microarrays which do not require any read alignments, sequence-based processing include trimming of unwanted bases from the reads, such as sequencing adapters or unwanted bases resulting from enzymatic end repair. Once that trimmed sequencing reads are aligned to the reference genome, methylation is called. Importantly, sequencing methods provide good resolution of genome-wide methylation, allowing to explore methylation patterns far beyond the single-site methylations shown by arrays (26).

A straightforward strategy to draw biological and functional consequences is to correlate DNAm with mRNA gene expression. In addition, plenty of computational tools are available, such as the R package BioMethyl which allows to integrate several algo-rithms for the interpretation of DNAm data (27). Another software, named EpiExplorer, provides interactive and live exploration of differentially methylated sites in the context of public reference epigenome data sets (28). The Galaxy software facilitates the comparison of data at genomic regions with other genomic data sets that are available online (29). Gene set enrichment analysis can be carried out using the GREAT Web server (30) which maps genomic regions to genes and controls the statistical analysis for the fact that genes differ in size and in their relative distance to each other. Interpretation of DNAm shifts integrated with other omics data, such as ChIP-seq, remains difficult, both technically and biologically.”

Reviewer 2 Report

The authors investigated the challenges in methylation-related psychiatric disorders. They have attempted to deal with the current knowledge and usage of methodological tools in analyzing epigenetics. They have addressed the issue of statistical robustness in the interpretation of epigenetic data.

I have enjoyed reading the present review. The topic is very interesting and the role of molecular mechanisms in psychiatric disorders is of major importance. Especially, the role of epigenetics is of outmost importance in psy-disorders. I would also suggest to the authors to add an older experiment, which described the role of breast-feeding in stress-related behavior. This topic is not studied at all and it is has been known to be crucial in human behavior.

In addition, I would suggest to the authors to add some info on miRNA regulation in psy-disorders, since their role is also significant in their regulation.

Finally, the authors should add a few more details on the interpretation of these data. NGS methylation-data are available in abundancy and their interpretation along with the available miRNA and mRNA expression data pose a challenge for their understanding. It is not necessary to go into depth with the algorithms available for such analyses, but it would be good to mention the available algorithms.

Overall, the present work is very interesting and it has merit for publication. I would encourage the authors to continue on this topic of research.

Author Response

I would also suggest to the authors to add an older experiment, which described the role of breast-feeding in stress-related behavior. This topic is not studied at all and it is has been known to be crucial in human behavior.

Thanks for this comment. We have now described this experiment in paragraph “1.2. DNAm as gene-environment interplay in depression and GAD”. “[…] Lastly, besides the beneficial effects of breastfeeding on the newborn infant, studies have reported that breastfeeding mothers report reductions in anxiety, negative mood, and stress when compared to formula-feeding mothers (31).”

“In addition, I would suggest to the authors to add some info on miRNA regulation in psy-disorders, since their role is also significant in their regulation.”

We thank the reviewer for this comment. We have added more information in paragraph “1.2. DNAm as gene-environment interplay in depression and GAD”. “[…] Importantly, DNAm may also affect the transcription of miRNA genes, which are part of the epigenetic regulatory network. miRNAs are short DNA transcribed molecules highly expressed in brain which modulate gene expression at the post-transcriptional level (32). In a recent review on post-mortem brain samples of depressed subjects, approximately 50 miRNAs were identified to be differentially expressed compared to healthy counterparts, however, miRNA-124, was consistently reported (see review (33)). In blood, miRNA-134 was significantly downregulated in depressed patients and in rats affected by mild stress (34). A study investigating miRNA levels in patients with GAD identified that circulating miRNA-4505 and miRNA-663 levels correlated with symptoms of anxiety (35). Lastly, miRNA-144 was found to have lower expression in patients with MDD and anxiety (36) and differential expression at cluster miRNA-17-92 was observed in mice with elevated anxiety- and depression-like behaviors (37).

Finally, the authors should add a few more details on the interpretation of these data. NGS methylation-data are available in abundancy and their interpretation along with the available miRNA and mRNA expression data pose a challenge for their understanding. It is not necessary to go into depth with the algorithms available for such analyses, but it would be good to mention the available algorithms.”

Many thanks for this comment. This aspect was also mentioned by Reviewer 1. We have now added this in the paragraph “1.3.3. Validation of the DNAm findings by a different technique or in an independent cohort”. Three approaches are well studied for DNAm interrogation, i.e. bisulfite conversion-based, methylation-sensitive-enzyme-restriction-based (MSRE) or affinity enrichment-based methods (23). During the MSRE and affinity-based approaches, DNA is not damaged by bisulfite treatment, however, they require more labor in retrieving the DNA fragments bound by the respective proteins. Furthermore, the interpretation of the DNAm data is somewhat challenging for this type of data. Following bisulfite conversion, the degree of DNAm is quantified by microarray or Next Generation Sequencing (NGS). By far, the most popular approach is the analysis of a preselected set of CpGs via DNA hybridization microarrays. […] In addition, reproducibility has been shown for different arrays, which makes it possible to compare previous results with new ones (24). […] The approaches based on bisulfite conversion do not distinguish between 5mC and 5-hydroxymethl cytosine modifications. This drawback is especially important in studies of DNAm in psychiatric disorders, as there is evidence for abundant tissue specific stable hyroxymethylation in neurons (25). In comparison to microarrays which do not require any read alignments, sequence-based processing include trimming of unwanted bases from the reads, such as sequencing adapters or unwanted bases resulting from enzymatic end repair. Once that trimmed sequencing reads are aligned to the reference genome, methylation is called. Importantly, sequencing methods provide good resolution of genome-wide methylation, allowing to explore methylation patterns far beyond the single-site methylations shown by arrays (26).

A straightforward strategy to draw biological and functional consequences is to correlate DNAm with mRNA gene expression. In addition, plenty of computational tools are available, such as the R package BioMethyl which allows to integrate several algo-rithms for the interpretation of DNAm data (27). Another software, named EpiExplorer, provides interactive and live exploration of differentially methylated sites in the context of public reference epigenome data sets (28). The Galaxy software facilitates the comparison of data at genomic regions with other genomic data sets that are available online (29). Gene set enrichment analysis can be carried out using the GREAT Web server (30) which maps genomic regions to genes and controls the statistical analysis for the fact that genes differ in size and in their relative distance to each other. Interpretation of DNAm shifts integrated with other omics data, such as ChIP-seq, remains difficult, both technically and biologically.”

Round 2

Reviewer 1 Report

The authors have addressed my concerns. I have no further comments or suggestions for them.